# Robotic Surgery for Non-Small Cell Lung Cancer Treatment in High-Risk Patients

**DOI:** 10.3390/jcm10194408

**Published:** 2021-09-26

**Authors:** Carmelina Cristina Zirafa, Gaetano Romano, Elisa Sicolo, Claudia Cariello, Riccardo Morganti, Lucia Conoscenti, Teresa Hung-Key, Federico Davini, Franca Melfi

**Affiliations:** 1Minimally Invasive and Robotic Thoracic Surgery, Robotic Multispecialty Center of Surgery, University Hospital of Pisa, 56124 Pisa, Italy; gaetano_romano1986@hotmail.com (G.R.); elisa.siki@hotmail.it (E.S.); lucia.conoscenti@gmail.com (L.C.); t_hungkey@yahoo.co.uk (T.H.-K.); davinifederico@gmail.com (F.D.); franca.melfi@unipi.it (F.M.); 2Cardiothoracic and Vascular Anaesthesia and Intensive Care, Department of Anaesthesia and Critical Care Medicine, University Hospital of Pisa, 56124 Pisa, Italy; c.cariello73@gmail.com; 3Section of Statistics, University Hospital of Pisa, 56124 Pisa, Italy; r.morganti@ao-pisa.toscana.it

**Keywords:** robotic lung surgery, ASA score, high-risk patients

## Abstract

Robotic-assisted pulmonary resection has greatly increased over the last few years, yet data on the application of robotic surgery in high-risk patients are still lacking. The objective of this study is to evaluate the perioperative outcomes in ASA III-IV patients who underwent robotic-assisted lung resection for NSCLC. Between January 2010 and December 2017, we retrospectively collected the data of 148 high-risk patients who underwent lung resection for NSCLC via a robotic approach at our institution. For this study, the prediction of operative risk was based on the ASA-PS score, considering patients in ASA III and IV classes as high-risk patients: of the 148 high-risk patients identified, 146 patients were classified as ASA III (44.8%) and two as ASA IV (0.2%). Possible prognostic factors were also analysed. The average hospital stay was 6 days (8–30). Post-operative complications were observed in 87 (58.8%) patients. Patients with moderate/severe COPD developed in 33 (80.5%) cases post-operative complications, while elderly patients in 25 (55%) cases, with a greater incidence of high-grade complications. No difference was observed when comparing the data of obese and non-obese patients. Robotic surgery appears to be associated with satisfying post-operative results in ASA III-IV patients. Both marginal respiratory function and advanced age represent negative prognostic factors. Due to its safety and efficacy, robotic surgery can be considered the treatment of choice in high-risk patients.

## 1. Introduction

Lung cancer is the leading cause of cancer-related mortality in the world, being the most commonly diagnosed malignancy worldwide. The World Health Organization (WHO) estimated, only in the year 2020, 2,206,771 cases of new diagnoses of lung tumours and 1,796,144 cases of death, meaning 11.7% and 18% of total diagnoses and death for cancer, for both sexes [1]. In Europe, lung cancer is the fourth most diagnosed type of tumour, following breast and prostate malignancies, accounting for 477,534 new cases per year. Moreover, lung neoplasm is the first cause of death by cancer in Europe, with 384,176 deaths just in 2020 [2].

The treatment of choice for the early stage and selected cases IIIA of non-small cell lung cancer (NSCLC) is represented by surgery. During the pre-operative phase, any patient fit for surgery should undergo pneumological, cardiological and anaesthesiologic evaluation [3]. An accurate preoperative patient risk assessment is to be considered a key step to achieve the best surgical outcome possible; hence, during the patient evaluation, the American Society of Anesthesiologists physical status (ASA-PS) is an important tool used to stratify the peri-operative risks for the anaesthesiologic assessment. In fact, ASA-PS can be considered an independent predictor for morbidity in patients who underwent anatomical pulmonary resection, and its use is also suggested to stratify the postoperative risk of complications [4].

Over the last few years, the increasing diffusion of the mini-invasive approach for lung cancer treatment has allowed the extension of its application to more advanced stages and increasingly complex cases. This greater application is undoubtedly consequent not only to the further development of the technology available, but also to the increased experience acquired by the operators with a consequent improvement in skills [5,6,7,8]. It has been widely demonstrated that less invasive approaches are associated with a faster recovery, less post-operative pain and a shorter period of hospitalization, while still maintaining the positive results obtained from the conventional approach in terms of oncological outcomes [9,10,11].

Robotic surgery, thanks to its advanced technology offering a set of features, such the three-dimensional vision, the tenfold magnified video imaging, the same degrees of freedom of the human hand displayed by robotic instruments, provides the same advantages of traditional mini-invasive approach, such as less traumatism and pain, shorter postoperative hospitalization and fewer postoperative complications [12]. Furthermore, the robotic approach grants intraoperatively a more comfortable and precise dissection of the hilar elements and lymphadenectomy, resulting in less hematic loss, and a safer movement into the thoracic cavity, thanks to the better manoeuvrability of robotic arms. Although robotic surgery during its early application was applied only to carefully selected clinically good condition early-stage NSCLC patients, gradually, over the years, the application of robotic thoracic surgery has been extended not only to complex cases but also in high-risk patients.

Our study aims at analysing the effectiveness and the safety of robotic pulmonary surgery for NSCLC in frail patients, characterized by a high peri-operative risk according to the ASA-PS score. 

## 2. Materials and Methods

We retrospectively collected and analysed the data of 326 consecutive patients who underwent robotic lung resection for NSCLC from January 2010 to December 2017 at the Robotic Surgical Centre of the University Hospital of Pisa. 

The patients underwent preoperative examination with computed tomography (CT) scan, positron emission tomography (PET) and cardiorespiratory evaluation. 

All surgical procedures were conducted by robotic surgery, with a full-endoscopic approach.

After discharge, patients underwent periodical outpatient follow-up with the evaluation of chest–abdomen CT scan or PET, according to medical evaluation, plus serum tumour markers dosage.

For this study, pre-operative, intra-operative and post-operative data were collected and analysed. The pre-operative data included in the study were: demographical characteristics, body mass index (BMI), smoking habits and clinical features of the patients (pulmonary function tests, comorbidities, evaluation of ASA-PS score). In this series, no patient underwent neoadjuvant chemotherapy.

The intra-operative data were: type of parenchymal resection, intraoperative complications and conversion. The postoperative data included: days of hospitalization, complications and related redo-surgery, hospital re-admission within 30 days from discharge and pathological analysis. In addition, long-term post-operative results such as overall survival were analysed. 

The ASA-PS score was calculated to identify high-risk patients by stratifying the perioperative risk. This classification is used to estimate the patient’s general health condition analysing their systemic pathologies, to stratify postoperative morbidity and mortality. Assigning a physical status classification level is a clinical decision based on multiple factors, among them the age of the patient, the BMI, smoking habit, alcohol consumption, pregnancy, diabetes, systemic hypertension, chronic obstructive pulmonary disease (COPD), ischemic heart disease, heart failure, patients with pacemaker, renal failure and dialysis therapy [13,14,15]. The ASA-PS classification system comprehends 6 classes of risk (Table 1). According to this score, patients were divided into two groups: patients with low perioperative risk (ASA-PS I-II) and patients at high risk (ASA-PS III-IV). Patients with ASA-PS V-VI were not considered as candidates for elective surgery and, hence, were not included in the study. 

Postoperative complications were stratified by using the Common Terminology Criteria for Adverse Events (CTCAE) grade; thus, higher scores are related to more severe postoperative complications, with grade 5 associated with death events [16].

### 2.1. Surgical Approach

General anaesthesia, with double-lumen intubation, was required during robotic lung surgery, for better management of the single lung ventilation. 

The patient is positioned on lateral decubitus with the operating table flexed at the scapula tip level. A totally endoscopic approach requires only four-port incisions for the procedure (Figure 1). The sites of surgical incisions are:-Seventh or 8th intercostal space, at the point where an imaginary line from the head of the humerus intersects the intercostal space. This first port hosts the 30-degree scope camera.-Seventh or 8th intercostal space posteriorly to camera port (about 8 cm).-Sixth or 7th intercostal space, in the auscultatory triangle area.-Sixth intercostal space on the anterior axillary line, above diaphragm insertion.

During the procedure, carbon dioxide insufflation was applied into the chest cavity, with a pressure ranging between 5 and 7 mmHg, to favour the collapse of the lung and the lowering of the diaphragm to maximize operative space.

### 2.2. Statistical Analysis

Data analysis was performed at the Robotic Minimally Invasive Thoracic Surgery Unit, University Hospital of Pisa, Italy. Categorical data were described by absolute and relative frequency, continuous data by mean and standard deviation. Survival curves were calculated using the Kaplan–Meier method, and the log-rank test was used to evaluate the differences between curves. To analyse OS risk factors, a multivariate Cox model was performed, and the results of the Cox regression were expressed by *p*-value, hazard ratio with 95% confidence interval and by regression coefficient. Significance was fixed at 0.05, and all analyses were carried out with SPSS Inc. version 27 (IBM SPSS Statistics for Windows, Armonk, NY, USA).

## 3. Results 

We retrospective analysed the data of 326 patients who underwent pulmonary resection with lymphadenectomy for NSCLC using the robotic approach, at our institution, from January 2010 to December 2017. The sample comprised 191 males (59%) and 135 females (41%), with a median age of 68.5 years (range 30–86). For each patient the ASA-PS score was calculated, resulting in five ASA-I (1.5%), 173 ASA-II (53.1%), 146 ASA-III (44.8%) and two ASA-IV (0.6%). 

Stratifying the risk, patients with low perioperative risk (ASA-PS I–II) were 178 (54.6%), while high-risk patients (ASA-PS III–IV) were 148 (45.4%). 

Clinical characteristics of the high-risk group are summarized in Table 1. In this group, the average age was 70 years (SD 7), with 47 patients (31.8%) over 75 years. The patients were 100 (67.6%) men and 48 (32.4%) women. Mean BMI resulted in 27 (SD 4.7; range 18–44.2) and 34 patients (23%) scored a BMI ≥ 30, while four patients (2.7%) presented severe obesity (BMI ≥ 40). One hundred and thirty-three patients (90%) were smokers, of which 25% were still regular smokers at operating time; only 15 patients (10%) never smoked in their life. Respiratory function tests showed median values of FEV1 86% (range 40–188), FVC 100% (range 45–189), FEV1/FVC 73% (range 31–98). 

The surgical and pathological details are summarized in Table 2. 

In nine cases (6.1%) conversion from robotic approach to thoracotomy was required. 

The median hospital stay was of 6 days (range 4–30). Postoperative complications were recorded in 87 patients (58.8%), and the most common was identified as being prolonged air leak, which occurred in 31 patients (20.9%). Details of postoperative complications are listed in Table 3.

According to CTCAE, adverse event occurred in 51 (56.7%) cases of grade 1, in 21 (23.3%) of grade 2, in 16 (17.8%) of grade 3 and in 2 (2.2%) of grade 5. In four (2.7%) cases, redo surgery due to prolonged air leak was performed. During the hospitalization, two patients died after cardiac complications. Only one patient (0.7%) was hospitalized within 30 days from discharge with a diagnosis of pneumothorax.

After a median follow-up time of 32 months (range 1–92), the actuarial OS was 84% at 12 months, 76% at 24 months, 67% at 36 months and 50% at 60 months (Figure 2).

According to the pathological stage, OS was 91% for early stages and 65% for advanced stages at 12 months, reaching 57% for early stages and 40% for advanced stages at 60 months, with a statistically significant difference (*p* = 0.011) (Figure 3). Twenty-one patients were lost at follow up.

Furthermore, in the analysis of results, we considered several different aspects in order to detect possible risk factors:SexThe high-risk group was composed of 48 (32%) women and 100 (68%) men. The median age was 71 years (range 53–83) for the women and 72 (range 48–86) for the men. No significant statistical difference was observed in the analysis of median hospital stay and complications rate.Elderly age (≥75 years)In the high-risk group, we observed a higher presence of elderly patients. In fact, 47 patients (31.8%) with ASA-PS III-IV were older than 75 years. We further divided this high-risk group into two subgroups (≥75 years versus <75 years) to evaluate the impact of older age on postoperative outcomes. The patients older than 75 were 47 (31.8%), with a median age of 77 years (range 75–86); whereas 101 (68.2%) patients were younger than 75 years, with a median age of 68 years (range 48–74). The median hospital stay was 7 days (range 4–30) for the ≥75 years patients and 6 days (range 4–26) for the <75 years group. The older patients developed postoperative adverse events in 25 (55%) cases, with a higher incidence of severe complications (25% CTCAE grade 1, 15% grade 2, 11% grade 3, 4% grade 5). The younger patients presented postoperative complications in 61 (60%) cases, with a lower incidence of severe adverse events (32% CTCAE grade 1, 20% grade 2, 8% grade 3). Obesity (BMI > 30)In the high-risk group, we identified three categories according to the BMI index: 50 (33%) patients presented a normal weight (BMI range 18–24.9), 64 (43%) were overweight (BMI range 25–29.9), and 34 (23%) were obese (BMI range >30). The median length of stay of overweight and obese patients was of 6 days (range 4–30), while for the normal-weight patients, this was 7 days (range 4–23). No statistically significant difference was observed in the evaluation of post-operative complications, albeit obese patients were characterized by less severe complications (Table 4).Smokers vs. non-smokersOne hundred and thirty-three (90%) of the high-risk patients were former or current smokers, while 15 (10%) never smoked. Postoperative complications occurred in 59% of smokers and 53% of no-smokers, with similar distribution in terms of severity according to CTCAE classification; the median length of stay was 7 (4–30) days in smoker patients and 6 (4–20) in non-smokers.◦Moderate or severe chronic obstructive pulmonary disease (FEV1/FVC <70%, FEV1 <80%)Spirometry of 41 (27.7%) patients showed an FEV1/FVC <70% associated with FEV1 <80%, diagnostic for moderate or severe COPD. A median hospital stay of 10 days (range 4–30) was observed in patients with moderate or severe COPD and of 6 days (range 4–26) in the other patients. Patients with moderate or severe COPD presented a higher rate of postoperative complications, with higher CTCAE grades. In addition, in this group of patients, prolonged air leaks were recorded in 54.5% of cases, in contrast to the rate of 24.1% observed in the other patients (Table 5).

When the impact of the possible risk factors on long-term outcomes was analysed, in the multivariate analysis of OS, the elderly age appears the only factor influencing survival, with a statistically significant difference (HR (95% CI) = 1057 (1014–1101); *p* value = 0.008).

## 4. Discussion

The use of the robotic approach for thoracic surgery has grown exponentially over the last decade both in the number of cases and in the complexity of cases treated such as in frail and high-risk patients. It has been proven over the years to be a feasible and valuable approach due to the advantages it provides such as low postoperative complication rate, reduced length of stay and post-operative pain, with satisfying aesthetic results. In fact, in the latest years, thanks also to the recent technology improvements, its application within the thoracic field has widely increased worldwide [17]. Technical features of the robotic system that can provide real advantages are stereoscopic binocular vision with highly magnified video 3D imaging and the 7 degrees of freedom characterizing the movement of robotic instruments, which transfers the surgeon’s movements to finer ones. The literature reports how a robotic approach to lung surgery can be considered safe, both in a surgical and an oncological point of view [18].

Thanks to the encouraging results obtained by using the robotic-assisted approach in thoracic surgery, it has been possible to extend the indications for this approach also to more challenging procedures [19]. Meanwhile, the trend of increasing life expectancy makes it necessary to offer the safest surgical approach possible, preferably in association with minimum traumatism, to patients considered frailer, being featured with high perioperative risk. To define the perioperative risk, the most used grading system is the ASA-PS score; it is widely applied to evaluate pre-operative health and the risk of perioperative complications in all surgical fields, including the thoracic surgery [20].

For these reasons, to select the group of “high-risk patients” who underwent robotic surgery for this analysis, we assumed its definition according to the ASA-PS classification.

Our analysis of the postoperative complications after robotic pulmonary resections showed similar results to those reported by other studies of patients who underwent conventional lung surgery, suggesting the possibility to also achieve positive postoperative outcomes when applying robotic surgery in high-risk patients. Conversely, the most common complication observed in our series was prolonged air leak with a higher incidence, occurring in 20% of patients [21]. The higher number of smoker patients and with a diagnosis of COPD, representing the main risk factors to develop prolonged air leak after lung resection, which could explain the higher incidence of these complications in our series [22,23,24]. In addition, the prolonged air leak incidence was found to be consistently reduced in patients who underwent surgery in recent years, taken into account in the study; this could be associated with the use of the last generation of the robotic system now available at our institution, which is characterized by higher technology and greater precision of movements.

When evaluating possible risk factors, we examined different characteristics of the sample to identify negative prognostic factors, affecting the postoperative outcomes of high-risk patients treated with minimally invasive lung resection

Respiratory functional status is known to be fundamental for the assessment of patients who are possible candidates for lung surgery [25]. In particular, FEV1 is one of the most commonly used parameters to estimates respiratory function, although its real predictive effectiveness to predict postoperative complications is still controversial [26,27]. To study the possible prognostic role of spirometry test values, we compared the postoperative results of patients with moderate and severe COPD with the results of the other high-risk patients. When comparing the two groups, a longer length of stay and a greater rate of postoperative complications, substantially consisting of major adverse events, were observed in those patients with a reduction in respiratory function. The patients with moderate or severe COPD were characterized by a higher rate of prolonged air leaks, and this percentage increases with the decreasing FEV1/FVC value, being 44% in COPD patients with FEV1/FEVC <70% and achieving 77% in patients with FEV1/FVC <50%.

Nevertheless, due to the less traumatism, higher technology and precision, the results achieved by the robotic approach in lung cancer treatment can be associated with fewer incidences of post-operative complications and faster recovery, also when applied in marginal pulmonary function patients [28].

Moreover, as has also been described by previous studies, we observed that smoker patients presented a higher frequency of respiratory complications, such as prolonged air leak, pneumoniae, empyema, respiratory failure, pulmonary embolism and pneumothorax, with a higher severity index (CTCAE grade of 2, 3 and 5). Nevertheless, the hospital stay and the rate of overall postoperative complications were similar when compared to the never-smoked patients, hinting at the peculiar association of smoking habit with higher risk of developing pulmonary complications.

Age is one of the necessary standards included in the assessment of the ASA-PS score, undoubtedly representing an essential aspect to be considered for the stratification of the surgical risk.

Recently, the increase in the average age of the population, in parallel with the development of more advanced surgical techniques available and the improvement of the mini-invasive approach, has allowed us to also consider surgical resection in the elderly patients, in which alternative treatments are often contraindicated. We observed a similar length of hospital stay and rate of post-operative complications in the elderly when compared to younger patients, albeit the adverse events of the elderly group occurred with higher CTACE grades.

The postoperative complications appear to negatively influence the surgical outcomes, both in the short- and long-term period. In fact, in the examination of long-term outcomes in high-risk patients, we observed that worse overall survival results were associated with increased age. Wang et al. have reported similar results, showing an association between major respiratory postoperative complications and worst long-term outcomes with higher mortality, not influenced by the lung cancer pathology. In detail, the independent risk factors affecting the long-term results were the ASA score, age and male sex [29].

Furthermore, our results showed that, in older patients, minor resections, segmentectomy and wedge resections were better tolerated, in terms of postoperative complications, than lobectomies. Indeed, the postoperative adverse events rate was 20% in patients who underwent minor resections, with a lower score of CTCAE grade, while they occurred in 60% of cases after lobectomies. Fewer post-operative complications were recorded after wedge resection. These data confirm that wedge resection can be considered a valuable option in the elderly population and in general in frail patients, burdened by various comorbidities or with a reduced respiratory function. Thus, when possible, applying surgery sparing lung parenchyma could reduce the onset of major post-operative complications in high-risk patients, with acceptable oncological outcomes [30].

According to several published studies, obesity does not constitute a risk factor in frail patients. On the contrary, obesity even seems to be a protective factor, being associated with lower length of stay and less severe postoperative complications, confirming the “obesity paradox” first described in 1999 [31]. Furthermore, no significant differences in the postoperative results were observed between the two sexes in the high-risk population, contrarily to other studies’ findings, showing lower postoperative morbidity and mortality after lung cancer surgery in women, associated with better survival than men. These results are probably related to the removal of common factors related to a survival advantage in women, such as age, comorbidities, physical performance, type and extent of surgery, tumour characteristics and stage of the disease, due to the selection of ASA PS score III–IV patients. [32,33].

Furthermore, the overall survival rate of frail patients observed in this study seems to be in line with the data reported in the literature for the general population, confirming the safety and efficacy of lung resection performed by robotic approach [34].

## 5. Conclusions

For high-risk NSCLC patients, in which often alternative treatment is not feasible, lung resection by robotic approach can represent a safe therapeutic option, in terms of short-term postoperative results and oncological results.

Nevertheless, a limited amount of data is present in the literature concerning the high-risk subjects who underwent robotic surgery to treat lung cancer; ergo, further study needs to be conducted.

## Figures and Tables

**Figure 1 jcm-10-04408-f001:**
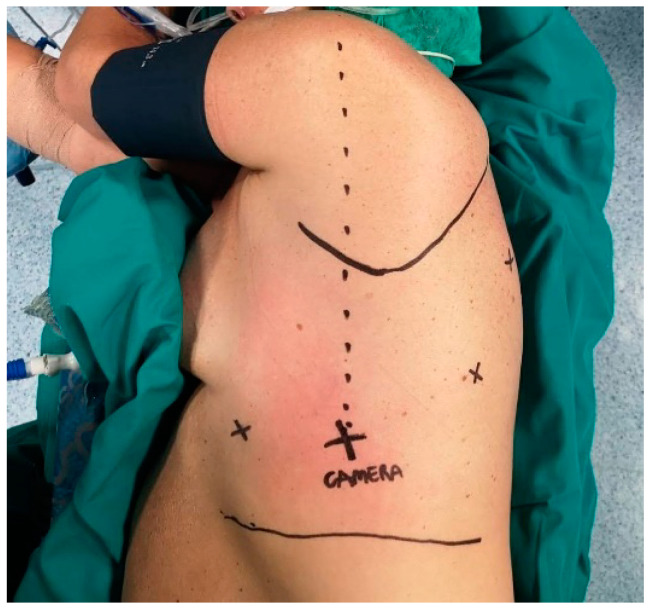
Standard port-mapping for totally endoscopic robotic lobectomy.

**Figure 2 jcm-10-04408-f002:**
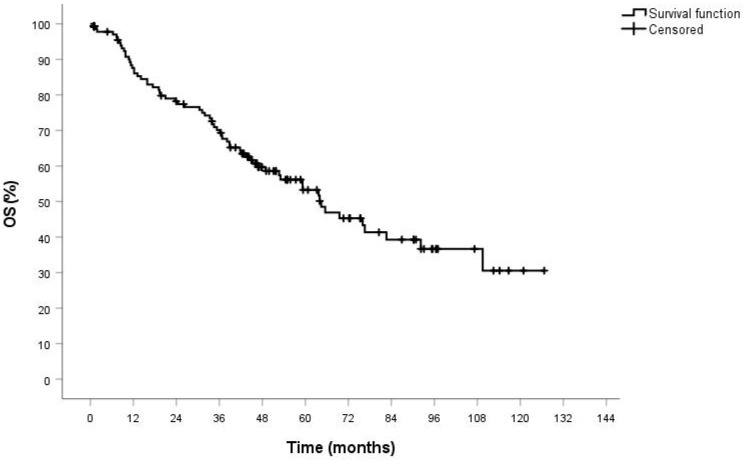
Overall survival of high-risk population after robotic surgery for lung cancer.

**Figure 3 jcm-10-04408-f003:**
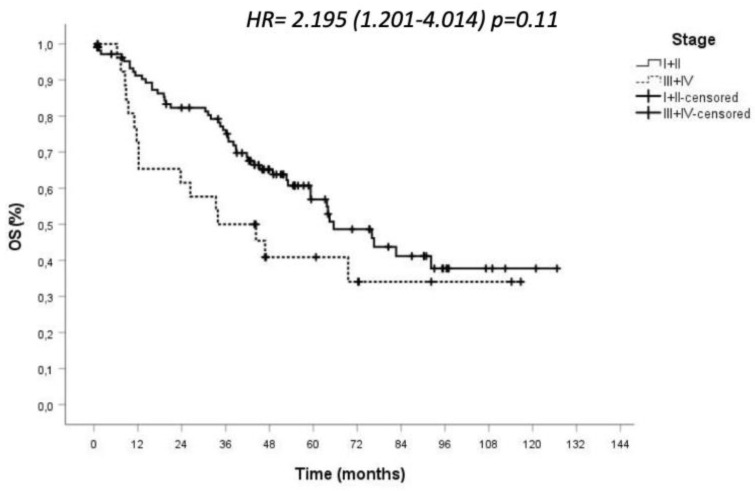
Overall survival of high-risk NSCLC patients according to tumour stage.

**Table 1 jcm-10-04408-t001:** ASA Physical Status Classification System.

ASA Score	Definition
ASA I	Healthy patients, non-smoking, no or minimal alcohol use
ASA II	Patients with mild systemic disease, without substantive functional limitations as current smokers, social alcohol drinkers, etc.
ASA III	Patients with severe systemic disease or substantive functional limitations. E.g., poorly controlled DM or HTN, COPD, active hepatitis, CVA, TIA or CAD/stents, etc.
ASA IV	Patients with severe systemic disease that is a constant threat to life. Examples: recent (<3 months) MI, CVA, TIA or CAD/stents, ongoing cardiac ischemia or severe valve dysfunction, severe reduction in ejection fraction, sepsis, etc.
ASA V	Moribund patients which are not expected to survive without the operation. Examples: ruptured abdominal/thoracic aneurysm, massive trauma, etc.
ASA VI	Declared brain-dead patients whose organs are being removed for donor purposes.

**Table 2 jcm-10-04408-t002:** Surgical and pathological characteristics of the population (*n* = 148).

Surgical Operation		pT Stage	
Bilobectomy	1 (0.7%)	T1	55 (37.2%)
Lobectomy RUL ML RLL LUL LLL	127 (85.7%) 43 (33.8%) 10 (7.9%) 25 (19.7%) 24 (18.9%) 25 (19.7%)	T2	66 (44.6%)
		T3	25 (16.9%)
		T4	2 (1.3%)
		**N Stage**	
Segmentectomy	10 (6.8%)	N0	111 (75%)
Wedge resection	10 (6.8%)	N1	16 (10.8%)
		N2	21 (14.2%)
**Histotype**		**pM Stage**	
Adenocarcinoma	96 (64.9%)	M0	147 (99.3%)
Squamous	30 (20.2%)	M1	1 (0.7%)
Neuroendocrine	14 (9.4%)	**Pathological Stage**	
Others	8 (5.5%)	I + II	119 (80.2%)
		III + IV	29 (19.8%)

**Table 3 jcm-10-04408-t003:** Post-operative complications after robotic lung resection for lung cancer in high-risk patients.

Post-Operative Complications	
None	61 (4.2%)
Prolonged air leak	31 (20.9%)
Subcutaneous emphysema	20 (13.5%)
Pleural effusion	10 (6.7%)
Atrial fibrillation	9 (6.1%)
Anemia	4 (2.7%)
Hypoxemia	4 (2.7%)
Type 1 respiratory failure	4 (2.7%)
Incomplete pulmonary re-expansion	4 (2.7%)
Recurrent laryngeal nerve palsy	3 (2%)
Hypokalaemia	3 (2%)
Hematoma	3 (2%)
Pneumothorax	3 (2%)
Heart failure	2 (1.4%)
Atelectasis	2 (1.4%)
Pulmonary thromboembolism	1 (0.6%)
Acute kidney injury	1 (0.6%)
Empyema	1 (0.6%)
Pneumoniae	1 (0.6%)
Diaphragm paralysis	1 (0.6%)

**Table 4 jcm-10-04408-t004:** Postoperative results according to the BMI class after lung resection by robotic surgery.

	Obese Patients *n* = 34 (23%)	Overweight Patients *n* = 64 (43.2%)	Normal Weight Patients *n* = 50 (33.8%)
BMI (median)	32.2 (30–44.2) Obese Patients Grade 1: 26 (76%) Grade 2: 4 (12%) Grade 3: 4 (12%)	27.1 (25–29.7)	22.9 (18–24.8)
Post-operative stay (median)	6 days (4–23)	6 days (4–30)	7 days (4–23)
Post-operative complications	In 21 (62%) patients: 13 AE Grade 1 5 AE Grade 2 3 AE Grade 3	In 34 (53%) patients: 18 AE Grade 1 10 AE Grade 2 8 AE Grade 3 1 AE Grade 5	In 32 (64%) patients: 20 AE Grade 1 6 AE Grade 2 5 AE Grade 3 1 AE Grade 5

**Table 5 jcm-10-04408-t005:** Postoperative outcomes according to respiratory function after robotic pulmonary resections.

	Moderate/Severe COPD *n* = 41 (27.7%)	Others *n* = 107 (72.4%)
FEV1 (range)	63% (40–79)	89% (49–188)
FVC (range)	92% (57–141)	105% (45–189)
FEV1/FVC (range)	54% (31–69)	74% (56–98)
Length of stay (range)	10 days (4–30)	6 days (4–23)
Post-operative complications	In 33 (80.5%) patients:	In 54 (46.7%) patients:
	36 AE	54 AE
	— 19 (52.7%) Grade I	— 32 (59.2%) Grade I
	— 6 (16.7%) Grade 2	— 15 (27.7%) Grade 2
	— 10 (27.8%) Grade 3	— 6 (11.1%) Grade 3
	— 1 (2.7%) Grade 5	— 1 (1.8%) Grade 5
	18 (54.5%) Prolonged air leak	13 (24.1%) Prolonged air leak
	3 (9%) Hypoxemia	7 (13%) Atrial fibrillation
	2 (6%) Atrial fibrillation	2 (3.7%) Anemia
	2 (6%) Anemia	2 (3.7%) Respiratory failure
	2 (6%) Respiratory failure	1 (1.8%) Hypoxemia
	9 (27.3%) Other	29 (53.7%) Other

## Data Availability

The data presented in this study are available on request from the corresponding author.

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
