# Peer review of "Robotic Surgery for Non-Small Cell Lung Cancer Treatment in High-Risk Patients"

_jcm, 2021, doi:10.3390/jcm10194408_

Round 1

Reviewer 1 Report

I think some literature is missing e.g. Sandhaus et al from 2020 in the Austrian Journal and the experinece from the book "Was gibt es Neues in der Chirurgie" from the same workgroup with Cheufou, Steinert and Sandhaus is also missing. I think this german experience is missing in the discussion.

Author Response

Thank you for your comments.

I kindly ask you to provide precise bibliographical references of the Sandhaus’s article which you cited. We were not able to find any Sandhaus’s article of 2020 related to our topic.  In addition, the book you cited is only available in German.

Reviewer 2 Report

The authors tried to demonstrate the applicability of robotic surgery to high-risk patients. However, the research is just the description of a single-institutional experience without any comparative design using control group, and no generalized findings can be obtained from this not well design. Furthermore, long duration of the surgery itself is usually contraindicated for high-risk patients. The analyses and interpretation of the data and inference from the sample are not sound.

The authors should scientifically evaluate the benefit of robotic surgery in terms of operation time, amount of blood loss, postoperative pain score, and so on. Postoperative complications and survival outcome should also be compared with the other approaches of VATS and open chest surgery. No scientific recommendations can be inferred from this small sample size and not well study design.

Author Response

Thank you for your kind comment.

Our study is based on a preliminary monocentric experience. The results of our research demonstrated that robotic surgery is feasible in selected high-risk patients, despite a mean operating time of about 200 min, we generally observed a fast recovery without compromising the post-operative outcomes. The comparison with other approaches will be certainly the subject of further study.

The reviewer suggested evaluating the benefit of robotic surgery (blood loss, post-operative pain…). This is not the focus of our study considering that previous studies have already confirmed that minimally invasive approaches, like robotic surgery, are associated with lower postoperative pain, fast recovery, reduced postoperative complications and positive oncologic outcomes despite the longer duration of surgical procedure.

Park BJ. Robotic lobectomy for non-small cell lung cancer: long-term oncologic results. Thorac Surg Clin. 2014 May;24(2):157-62, vi. doi: 10.1016/j.thorsurg.2014.02.011. PMID: 24780419.

Wei B, Eldaif SM, Cerfolio RJ. Robotic Lung Resection for Non-Small Cell Lung Cancer. Surg Oncol Clin N Am. 2016 Jul;25(3):515-31. doi: 10.1016/j.soc.2016.02.006. PMID: 27261913.

Abbas AE. Surgical Management of Lung Cancer: History, Evolution, and Modern Advances. Curr Oncol Rep. 2018 Nov 13;20(12):98. doi: 10.1007/s11912-018-0741-7. PMID: 30421260.

Reviewer 3 Report

I have read with an interest a manuscript entitled: ROBOTIC SURGERY FOR NON-SMALL CELL LUNG CANCER TREATMENT IN HIGH-RISK PATIENTS. The paper is interesting and it was a pleasure to review it. First of all, I would like to ask the authors to perform significant language editing as I find a number of clumsiness that make parts of the text difficult to understand. A large number of punctuation, grammatical and stylistic errors give the impression of carelessness. I have some remarks concerning the manuscript.

Major remarks.

Line 20. The authors decided to define the high-risk group as ASA III-IV patients. This is not a commonly chosen factor for lung cancer surgery. Please comment on this choice in the introduction and in the discussion. Commonly chosen factors that define a high-risk population in lung cancer surgery are different – for example, pulmonary function tests. The reader must remember that the ASA scale refers mainly to the safety of the narrowest perioperative period and the process of general anesthesia. The qualification in lung cancer surgery must go beyond in order to provide not only low 30/90-day mortality but also preserving the quality of life and feasibility to eventual adjuvant treatment. I do not reject the concept of the ASA scale. However, this choice deserves a wider discussion.

Line 82. In order to reduce selection bias please provide a study flowchart indicating how many patients were excluded from the analysis due to different reasons.

Line 86. Please state how many patients were after neoadjuvant treatment.

Table 3. Some of the complications (dysphonia, hypokalemia) are not commonly recognized. Please reconsider redefining these complications (recurrent nerve injure instead of dysphonia) or including them in a wider category – other.

Figure 3. The quality of the figure is low. Please provide numbers at risk at the bottom of the figure.

Lines 180-181. What were the methods of follow-up? How many patients were lost to follow-up?

Lines 239-241. What were the factors taken into univariate analysis before multivariate one? Please provide a table disclosing this analysis. Was the age taken as a categorical or continuous variable?

The study would clearly benefit from the comparison of high risk vs. low risk, robotic vs. VATS, robotic vs. open. I regret the authors did not challenge this approach.

Minor remarks.Line 106. Aren’t you mixing the Italian abbreviation with the English extension (COPD – BPCO)?

Author Response

Thank you for your precious suggestions.

Assuming that robotic thoracic surgery is rapidly spreading worldwide in the last years, it was observed that younger patients in good general conditions are preferred in the first phases of the learning curve and of its application. Subsequently, due to the increasing surgical and anesthesiological expertise, the indication can be extended also to more complex cases.

  • To define the perioperative risk, the most used grading system is the ASA-PS score. This score is widely applied to evaluate pre-operative health and the risk of peri-operative complications in all surgical fields. Also in thoracic surgery, the use of ASA score is a validated method to determine the perioperative risks. For these reasons, to select the group of “high-risk patients” who underwent robotic surgery for this analysis, we assumed its definition according to the ASA-PS classification. The other most common risk factor for lung surgery (poor respiratory function, old age…) were analyzed separately.

  • Considering the frailty characterizing the group of the patients analyzed, no neoadjuvant therapy was administered in these patients due to his high risk of complications, according to previous multidisciplinary evaluation.
  • We modified the definition of dysphonia as you suggested. In contrast, Hypokalemia is a commonly recognized adverse event according to CTCAE.
  • As reported in the paper (line 90) patient was evaluated periodically by outpatient consultation
  • The analysis of data, considering the characteristics of the sample, was conducted only as a multivariate analysis. In addition, age was considered a continuous variable.
  • This is only a preliminary experience. Further studies are necessary to confirm the results obtained and also to compare low risk vs high-risk patients who underwent robotic surgery and compare surgical approaches.
  • “BPCO” in line 106 has been corrected as you suggested.
  • We modify table 3

Round 2

Reviewer 2 Report

This study is just the description of a single-institutional experience, and  I found little improvement in this revision. I think its clinical impact is quite limited.

Author Response

Thanks for your valuable observation.

Our study is a preliminary experience that confirms the feasibility of the robotic approach even in high-risk patients. We are part of a high volume center, which has allowed us to extend the surgical indication to patients who were previously excluded from minimally invasive surgery.

As you kindly suggested, we will involve other centers in the future to broaden our case series and confirm the results.

Reviewer 3 Report

I would like to thank the authors for their efforts. The paper is currently well written. However, the authors neglected to comment on a few issues mentioned by me:

Line 82. In order to reduce selection bias please provide a study flowchart indicating how many patients were excluded from the analysis due to different reasons.

Repeated remark - the study flow chart is careless. Please elaborate.

---

Line 86. Please state how many patients were after neoadjuvant treatment.

Repeated remark - I do not find this information.

---

Figure 3. The quality of the figure is low. Please provide numbers at risk at the bottom of the figure.

Repeated remark - no numbers at risk provided.

---

Lines 180-181. What were the methods of follow-up? How many patients were lost to follow-up?

Repeated remark - no information on this data in the text.

Author Response

  • We apologize for the inconvenience. We uploaded the correct flow chart file.
  • Information added in line 158, as you previously suggested
  • Thank you for your valuable suggestion. Unfortunately we are not able to further improve the image quality
  • Information added in line 152 and 304 as you suggested.

Round 3

Reviewer 2 Report

I can't think of robotic surgery as less invasive, considering its long duration of procedure. Thus, I think it is neither useful nor ethical to perform it in high-risk patients.

What is the difference between VATS and robotic surgery in terms of the surgical benefit to the high-risk patients?

Reviewer 3 Report

I would like to thank the authors for their efforts and improvements. I think that the paper still has some flawbacks, however may be published.